# Enhancing Vision-Language Model with Unmasked Token Alignment at Scale

## Abstract

Contrastive pre-training on image-text pairs, exemplified by CLIP, becomes a standard technique for learning multi-modal visual-language representations. Although CLIP has demonstrated remarkable performance, training it from scratch on noisy web-scale datasets is computationally demanding. On the other hand, mask-then-predict pre-training approaches, like Masked Image Modeling (MIM), offer efficient self-supervised learning for single-modal representations. This paper introduces **U**nmasked **T**oken **A**lignment (UTA), a method that leverages existing CLIP models to further enhance its vision-language representations. UTA trains a Vision Transformer (ViT) by aligning unmasked visual tokens to the corresponding image tokens from a frozen CLIP vision encoder, which automatically aligns the ViT model with the CLIP text encoder. The pre-trained ViT can be directly applied for zero-shot evaluation even without training on image-text pairs. Compared to MIM approaches, UTA does not suffer from training-finetuning inconsistency and is much more training-efficient by avoiding using the extra [MASK] tokens. Extensive experimental results demonstrate that UTA can enhance CLIP models and outperform existing MIM methods on various uni- and multi-modal benchmarks.

## 1 Introduction

Contrastive pre-training, e.g., CLIP (Radford et al., 2021), with web-scale image-text pairs is becoming the mainstream technique for learning multi-modal visual-language representations. The pre-trained CLIP model has unlocked the potential of various downstream applications, including zero-shot image classification and retrieval, and high-quality text-to-image generation (Rombach et al., 2022; Ramesh et al., 2022). Furthermore, the pre-trained visual and text encoders can be further used for multi-modal and even uni-modal tasks.

Unlike classical supervised learning on the human-annotated classification dataset, CLIP and its variants are typically trained on much noisier datasets found on the web such as LAION (Schuhmann et al., 2022) and WIT (Radford et al., 2021), and require an extremely large batch size to work well. Directly training on those datasets from scratch requires a lot of computing resources, making it not accessible to most researchers. In contrast, the mask-then-predict pre-training approaches, e.g., Masked Image Modeling (MIM) (He et al., 2021; Xie et al., 2021) and Masked Language Modeling (MLM) (Devlin et al., 2019), have been shown to be efficient and powerful way to learn single-modal (visual or language) representations in self-supervised manner and can achieve strong performance by fine-tuning the pre-trained models on downstream tasks. The key design of those methods is to predict the masked tokens from the other visible and unmasked input tokens. We ask the question: can we take advantage of both types of methods and further enhance the vision-language representations over CLIP? There are recent works, e.g., EVA (Fang et al., 2023b), utilizing a pre-trained CLIP model for generating the prediction targets for MIM. The resulting vision models show stronger performance than the encoders pre-trained using either only MIM or only CLIP, demonstrating the effectiveness of combining MIM and CLIP for multi-modal feature learning. However, those methods are limited to learning single-modal representations, and extra contrastive fine-tuning is needed for multi-modal feature learning, as proposed in EVA-CLIP (Sun et al., 2023).

In this paper, we propose an efficient method, Unmasked Token Alignment (UTA), for enhancing the alignment between vision-language representations, which better utilizes existing pre-trained CLIP models. In particular, our method trains a Vision Transformer (ViT) (Dosovitskiy et al., 2021)

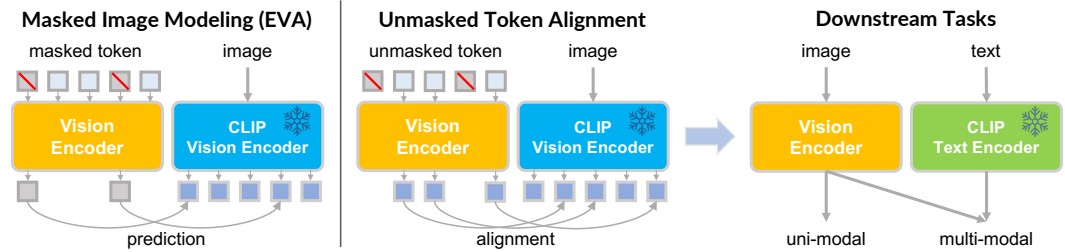

Figure 1: Overview of Unmasked Token Alignment (UTA). During the pre-training of UTA, only the unmasked tokens are inputted into the vision encoder and aligned with the CLIP vision encoder. After pre-training, the pre-trained vision encoder is automatically aligned with the CLIP text encoder and can be directly applied for the zero-shot evaluation even without contrastive training on image-text pairs. The pre-trained vision encoder can be further fine-tuned for uni-modal or multi-modal downstream tasks.

model from scratch by using the unmasked and sparse visual tokens to align with corresponding image tokens of a frozen CLIP model. For the train-from-scratch ViT model, we randomly mask a portion of image tokens with a *reversed* masking strategy, where only the unmasked (i.e. kept) tokens (including the [CLS] token) are inputted into the ViT model and aligned with the output of the frozen CLIP visual model. We maximize the cosine similarity for token alignment, and therefore, the ViT model is automatically aligned with the CLIP text encoder in the normalized embedding space.

There are two major advantages of using the proposed unmasked token alignment strategy. 1) After pre-training the vision model, we can directly conduct zero-shot classification and retrieval using the normalized features of the trained ViT model and the CLIP text encoder. We illustrate the pre-training and fine-tuning pipeline of UTA in Fig. 1. In contrast, the masked prediction objective used in existing MIM works (EVA (Fang et al., 2023b), BEiT-3 (Wang et al., 2022b)) relies on the [MASK] tokens to predict the CLIP features while the unmasked tokens are not trained to align with the CLIP model as we do. They do not support zero-shot evaluation without contrastive fine-tuning as only the unmasked tokens are used for zero-shot evaluation. 2) MIM works suffer from the training-finetuning inconsistency as a large portion of [MASK] tokens never appear during the fine-tuning. In contrast, our approach better maintains the training-finetuning consistency by only inputting and aligning the unmasked tokens, which are processed both in training and inference. We also empirically find that further adding the masked prediction objective on our UTA results in much worse zero-shot performance.

Compared to the existing MIM approach that relies on the [MASK] tokens to predict the CLIP features with the masked prediction objective, our method is conceptually simple and computationally efficient by avoiding introducing the [MASK] tokens, which can reduce the training FLOPs for up to 50%. But at the same time, our pre-trained models are also suitable for fine-tuning on downstream uni-modal or multi-modal tasks. In particular, our pre-trained ViT-L obtains 78.5% zero-shot accuracy on ImageNet without contrastive fine-tuning from image-text pairs. After fine-tuning with the DataComp-1B dataset (Gadre et al., 2023), we obtained 80.8% zero-shot accuracy on ImageNet, surpassing the DataComp baseline and EVA-CLIP by 1.6% and 1.0%, respectively. On the more recent multi-modal benchmark, i.e., LLaVA-Bench (Liu et al., 2023), we outperform CLIP and EVA-02 by 2.2% and 1.4%, respectively. We also fine-tune the pre-trained vision model on object detection and segmentation tasks and demonstrate better results than the competitive EVA-02 (Fang et al., 2023a) models on those tasks.

## 2 METHOD

In this section, we first review the widely used Masked Image Modeling (MIM) pre-training and its more advanced version equipped with a pre-trained CLIP model. We then introduce the unmasked token alignment (UTA) approach and its implementation.

## 2.1 A Revisit of Masked Image Modeling with CLIP

MIM methods (Bao et al., 2021; He et al., 2021; Xie et al., 2021) typically use a Vision Transformer (ViT) (Dosovitskiy et al., 2021) for pre-training. An input image is first divided into non-overlapping image patches, which are converted into a sequence of tokens with a project layer and positional embedding. Then a portion of the tokens are randomly sampled, where the masked tokens are filled with a special [MASK] token. The masked image is processed by the ViT to produce the latent representations, and a lightweight head is utilized to predict the original image based on the latent representations. After pre-training, the ViT is used for further fine-tuning on downstream visual tasks.

Some recent papers (Peng et al., 2022; Fang et al., 2023b; Hou et al., 2022; Xiao et al., 2022) utilize the hidden features of a pre-trained CLIP model as the reconstruction targets and achieve much better performance than methods using the low-level pixels as the targets (He et al., 2021; Xie et al., 2021). In particular, the unmasked image is fed into the visual encoder of the CLIP model for obtaining the full image's hidden feature map. The masked prediction objective is to align the predicted feature with the CLIP's visual feature on the masked tokens.

## 2.2 Unmasked Token Alignment

Using the masked prediction objective to align a train-from-scratch ViT model with the pre-trained CLIP visual model still uses the problematic [MASK] tokens. It causes training-finetuning inconsistency and makes the trained ViT unable to perform zero-shot classification without fine-tuning. To tackle the issue, we propose a simple yet effective solution that does not utilize the extra [MASK] tokens. We align the feature maps of the two models with a dense distillation objective, where the feature maps of the train-from-scratch ViT model and CLIP vision encoder are obtained with a partial view and a full view, respectively. Specifically, given an input image, we use a random mask to mask a portion of image tokens. Unlike previous works that use the [MASK] tokens to fill in the masked patches, we directly drop the masked tokens and only input the rest tokens into the ViT encoder. For the pre-trained CLIP model, we input the original image and obtain a full hidden feature map. Then we select the corresponding unmasked (kept) tokens from the CLIP vision encoder's feature map, which are used as the targets for the train-from-scratch ViT encoder.

The cosine similarity is maximized for the token alignment. After pre-training, the ViT encoder is aligned with the CLIP vision encoder in the normalized embedding space. Therefore, the ViT encoder is also aligned with the CLIP text coder as the CLIP's vision and text encoders share the same embedding space. As a result, we can directly conduct the zero-shot evaluation with the pre-trained ViT encoder and CLIP text encoder even without training on the image-text pairs. We show that we can already achieve decent zero-shot performance after the unmasked alignment.

**Reversed block-wise masking.** Previous works (Bao et al., 2021) typically use block-wise masking to preserve the structure of input images. However, we note that such masking is spatially unequalized, which tends to mask the center area of the images with a much higher probability, and as a result, the tokens in the border area are trained much more times than tokens in the center area. We introduce a reversed block-wise masking strategy, which first generates a mask with block-wise masking and then randomly reverses the mask with a probability of 0.5. Our masking strategy preserves the structure of the input images and also alleviates the spatial unequalization problem.

**Pre-training efficiency analysis.** As we do not need to process the extra [MASK] tokens during the pre-training, we can largely improve the masked training efficiency. In practice, we use a large mask ratio, e.g., 0.5, for pre-training. Thus, compared to EVA (Fang et al., 2023b) or BEiT v2 (Peng et al., 2022) which require inputting extra [MASK] tokens, our UTA can reduce the training FLOPs by 50%.

## 2.3 Implementation

**Vision transformer architecture.** We follow EVA-02 (Fang et al., 2023a) to introduce architectural modifications on vision transformer for improving the performance and training stability. In particular, we add extra relative positional embedding introduced by Su et al. (2021) in the self-attention layer. We replace the original feedforward network (FFN) in vision transformer with the SwiGLU variant introduced by Shazeer (2020). Moreover, we add an extra LayerNorm (Ba et al., 2016) layer in the FFN to stabilize the training as proposed by Wang et al. (2022a).

**CLIP teacher model.** Instead of using original CLIP models for pre-training, we follow Fang et al. (2023a) to use a better-performing CLIP model, i.e., giant-sized EVA-CLIP model (Sun et al., 2023), for providing the alignment targets during pre-training. Our experiments show that the stronger CLIP model can bring large zero-shot accuracy improvements. Additionally, we find the pre-trained ViT-L model can surpass the giant-sized CLIP model after contrastive fine-tuning.

## 3    EXPERIMENTAL SETUP

To demonstrate the effectiveness of the proposed Unmasked Token Alignment (UTA), we conduct experiments to pre-train ViT to align with CLIP vision-language representation on large-scale dataset and apply the pre-trained models to downstream multi-modal and uni-modal tasks. The multi-modal tasks include zero-shot classification, zero-shot retrieval, and the more recent LLaVA-Bench (Liu et al., 2023). The uni-modal tasks include ImageNet classification (Deng et al., 2009), object detection, and segmentation.

**Pre-training.** All ViT models are pre-trained on ImageNet-21K (Deng et al., 2009) dataset using 224×224 input resolution. Unless otherwise specified, we pre-train for 150 epochs with batch size of 4096. We use AdamW (Loshchilov & Hutter, 2017) optimizer with weight decay of 0.05. The learning rate is linearly increased to $1.5 \times 10^{-3}$ with 1 epoch of training and decays to $10^{-5}$ with cosine schedule (Loshchilov & Hutter, 2016). By default, we use reversed block-wise masking with mask ratios of 0.4 and 0.5 for base and large models, respectively.

**Contrastive fine-tuning.** Although the pre-trained ViT model can already demonstrate excellent zero-shot capabilities even without contrastive fine-tuning, we also perform a much shorter contrastive fine-tuning similar to other CLIP counterparts to further improve its zero-shot performance, especially for the out-of-distribution tasks. In particular, we initialize the vision and text encoders with the pre-trained ViT model and CLIP text encoder. Then we perform contrastive fine-tuning on the DataComp-1B dataset (Gadre et al., 2023). The temperature parameter in the contrastive loss (Radford et al., 2021) is fixed to 0.01 during our training as initially the vision encoder and text encoder are already aligned.

**Fine-tuning.** For evaluation on the LLaVA-Bench (Liu et al., 2023) and uni-modal tasks, we only keep the pre-trained ViT. On LLaVA-Bench, we follow the default settings to first train a projection layer on CC-3M dataset (Sharma et al., 2018) for feature alignment and then fine-tune the project layer and Large Language Model (LLM) (Chiang et al., 2023) on LLaVA-Instruct-150K dataset (Liu et al., 2023). For object detection and instance segmentation tasks, we adopt the Cascade Mask R-CNN (He et al., 2017; Cai & Vasconcelos, 2019) framework and separately fine-tune on the COCO (Lin et al., 2014) and LVIS (Gupta et al., 2019) datasets. For semantic segmentation task, we adopt the UperNet (Xiao et al., 2018) framework and fine-tune on the ADE20K (Zhou et al., 2017) dataset. Please refer to the appendix A.1 for more detailed configurations.

## 4    MAIN RESULTS

In this section, we compare the proposed Unmasked Token Alignment (UTA) to prior arts on various benchmarks. We first conduct comparisons between UTA and previous zero-shot results in Sec. 4.1. We then compare UTA with other pre-training methods on LLaVA-Bench in Sec. 4.2. To show the transferability of UTA, we present the transfer learning results on core vision tasks in Sec. 4.3.

### 4.1    ZERO-SHOT RESULTS

We conduct zero-shot classification and retrieval and compare the results with other CLIP variants (Radford et al., 2021; Cherti et al., 2023; Sun et al., 2023). In Tab. 1, we show that the pre-trained ViT-B model can obtain 76.0% zero-shot accuracy on ImageNet-1K even without training on image-text pairs. After fine-tuning with only 2B image-text samples, our ViT-B obtains 77.0% zero-shot accuracy on ImageNet-1K, surpassing Open-CLIP (Cherti et al., 2023) and EVA-CLIP (Sun et al., 2023) by 2.3% and 1.0% respectively. On the challenging ObjectNet (Barbu et al., 2019) dataset, we outperform Open-CLIP and EVA-CLIP by 11.3% and 6.0% points respectively. Our pre-trained ViT-L model obtains 78.5% zero-shot accuracy on ImageNet-1K. After fine-tuning with 4B samples, we achieve 80.8% accuracy, which outperforms Open-CLIP and EVA-CLIP by 5.3% and 1.0%

Table 1: Zero-shot classification performance on ImageNet-1K (IN-1K), ImageNet-A (IN-A) (Hendrycks et al., 2021b), ImageNet-R (IN-R) (Hendrycks et al., 2021a), ImageNet-V2 (IN-V2) (Recht et al., 2019), ImageNet-Sketch (IN-S) (Wang et al., 2019), and ObjectNet (Barbu et al., 2019). We also report the average accuracy over the 6 datasets.

| Method | Model | # I-T Pairs | IN-1K | IN-A | IN-R | IN-V2 | IN-S | ObjectNet | Average |
|---|---|---|---|---|---|---|---|---|---|
| CLIP | B/16@224 | 13B | 68.3 | 50.0 | 77.7 | 61.9 | 48.2 | 55.3 | 60.2 |
| Open-CLIP | B/16@224 | 34B | 70.2 | 38.2 | 80.6 | 62.3 | 56.1 | 56.0 | 60.6 |
| EVA-02-CLIP | B/16@224 | 8B | 74.7 | 54.1 | 82.5 | 67.0 | 57.7 | 62.3 | 66.4 |
| UTA | B/14@224 | **0B** | 76.0 | 54.2 | 76.7 | 68.1 | 52.5 | 63.6 | 65.2 |
| UTA | B/16@224 | 2B | **77.0** | **59.8** | **84.1** | **69.5** | **60.2** | **68.3** | **69.8** |
| CLIP | L/14@224 | 13B | 74.0 | 48.0 | 86.5 | 66.4 | 61.8 | 61.1 | 66.3 |
| Open-CLIP | L/14@224 | 32B | 75.5 | 70.8 | 87.8 | 69.9 | 59.6 | 69.0 | 72.1 |
| DataComp | L/14@224 | 13B | 79.2 | 69.6 | 90.8 | 72.1 | 68.0 | 74.3 | 75.7 |
| EVA-02-CLIP | L/14@224 | 4B | 79.8 | 76.1 | **92.7** | 72.9 | 68.1 | 75.3 | 77.5 |
| UTA | L/14@224 | **0B** | 78.5 | 69.4 | 89.4 | 71.7 | 63.9 | 72.7 | 74.3 |
| UTA | L/14@224 | 4B | **80.8** | **79.1** | 92.3 | **73.7** | **68.4** | **77.6** | **78.6** |
| CLIP | L/14@336 | 13B | 76.6 | 77.5 | 89.0 | 70.9 | 61.0 | 72.0 | 74.5 |
| EVA-02-CLIP | L/14@336 | 6B | 80.4 | 82.9 | **93.2** | 73.8 | 68.9 | 78.4 | 79.6 |
| UTA | L/14@336 | **4B** | **81.4** | **84.2** | 92.9 | **74.6** | **69.1** | **80.1** | **80.4** |
| Open-CLIP | g/14@224 | 34B | 78.5 | 60.8 | 90.2 | 71.7 | 67.5 | 69.2 | 73.0 |
| EVA-01-CLIP | g/14@224 | 11B | 79.3 | 74.1 | 92.5 | 72.1 | 68.1 | 75.3 | 76.9 |
| UTA | g/14@224 | **0B** | 79.3 | 73.5 | 91.6 | 72.6 | 66.7 | 74.6 | 76.4 |
| UTA | g/14@224 | 2B | **81.5** | **81.9** | **93.5** | **74.8** | **69.6** | **79.7** | **80.2** |

Table 2: Zero-shot retrieval performance on Flickr30k (Young et al., 2014) and COCO (Lin et al., 2014). R@1, R@5, and R@10 denote the recall performance among top-1, top-5, and top-10, respectively.

| Method | Model | # I-T Pairs | Text Retrieval | | | | | | Image Retrieval | | | | | |
|---|---|---|---|---|---|---|---|---|---|---|---|---|---|---|
| | | | Flickr30k | | | COCO | | | Flickr30k | | | COCO | | |
| | | | R@1 | R@5 | R@10 | R@1 | R@5 | R@10 | R@1 | R@5 | R@10 | R@1 | R@5 | R@10 |
| CLIP | B | 13B | 81.9 | 96.2 | 98.8 | 52.4 | 76.8 | 84.7 | 62.1 | 85.6 | 91.8 | 33.1 | 58.4 | 69.0 |
| Open-CLIP | B | 34B | 86.3 | 97.9 | 99.4 | 59.4 | 81.8 | 88.6 | 69.8 | 90.4 | 94.6 | 42.3 | 66.7 | 77.1 |
| EVA-02-CLIP | B | 8B | 85.7 | 96.7 | 98.9 | 58.7 | 80.7 | 88.2 | 71.2 | 91.0 | 94.7 | 42.4 | 66.9 | 76.3 |
| UTA | B | **0B** | 88.4 | 98.5 | 99.5 | 63.4 | 83.9 | 90.0 | **75.5** | **93.1** | **96.4** | **46.8** | **71.5** | **80.8** |
| UTA | B | 2B | **91.3** | **98.9** | **99.7** | **64.7** | **85.0** | **90.5** | 74.5 | **93.1** | 96.0 | 45.9 | 70.5 | 79.3 |
| CLIP | L | 13B | 85.2 | 97.3 | 99.0 | 56.3 | 79.3 | 86.7 | 65.2 | 87.3 | 92.0 | 36.5 | 61.0 | 71.1 |
| Open-CLIP | L | 34B | 88.7 | 98.4 | 99.2 | 62.1 | 83.4 | 90.3 | 75.0 | 92.5 | 95.6 | 46.1 | 70.7 | 79.4 |
| EVA-02-CLIP | L | 4B | 89.7 | 98.6 | 99.2 | 63.7 | 84.3 | 90.4 | 77.3 | 93.6 | 96.8 | 47.5 | 71.2 | 79.7 |
| UTA | L | **0B** | 91.2 | 98.7 | **99.8** | **66.6** | 86.5 | 91.5 | **78.3** | 94.1 | 96.9 | 49.5 | 73.4 | 81.9 |
| UTA | L | 4B | **93.0** | **99.0** | 99.7 | 66.5 | **86.9** | **92.2** | 77.4 | 93.8 | 96.6 | 48.7 | 72.3 | 80.9 |
| Open-CLIP | g | 34B | 91.4 | 99.2 | 99.6 | 66.4 | 86.0 | 91.8 | 77.7 | 94.1 | 96.9 | 48.8 | 73.3 | 81.5 |
| EVA-01-CLIP | g | 11B | 91.6 | 99.3 | 99.8 | **68.2** | 87.5 | 92.5 | 78.9 | **94.5** | 96.9 | **50.3** | 74.0 | 82.1 |
| UTA | g | **0B** | 92.2 | 99.1 | 99.7 | 68.0 | 87.2 | 92.4 | **79.0** | **94.5** | **97.2** | **50.3** | 74.2 | **82.5** |
| UTA | g | 2B | **93.2** | **99.4** | **99.8** | **68.2** | **87.6** | **93.0** | 78.2 | 94.4 | 96.7 | 48.7 | 72.9 | 81.1 |

respectively. Compared to strong EVA-CLIP, we achieve an average of 1.1% improvements over 6 evaluation datasets. We also fine-tune with 336×336 input resolution using 200M samples, and we obtain an average of 1.8% points improvements on the 6 evaluation datasets. We find that fine-tuning on the larger but noisier DataComp-1B dataset (Gadre et al., 2023) can greatly boost the performance on the ImageNet robust variants.

Table 2 presents the zero-shot retrieval results on the Flickr30k (Young et al., 2014) and COCO (Lin et al., 2014) datasets. We find that the pre-trained model can already outperform other CLIP models on all evaluated metrics. In particular, the base model improves the Open-CLIP and EVA-CLIP by an average of 4% top-1 recall over the two datasets. For the large model, we improve the Open-CLIP and EVA-CLIP by an average of 3.4% and 1.8% top-1 recall, respectively. We also find that further fine-tuning on DataComp-1B dataset can improve the text retrieval performance but also degenerate the image retrieval performance.

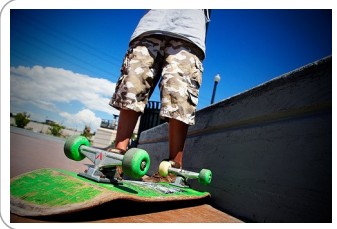

**Question: What is the position of the skateboard in the image?**
**EVA:** The skateboard is on the ground, with the person standing on top of it.
**UTA:** The skateboard is positioned upright, with the wheels off the ground, and the deck facing upwards.

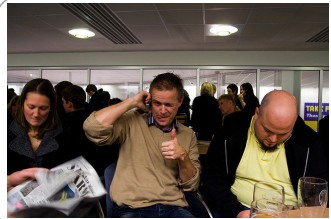

**Question: What is the man sitting in the middle doing in the image?**
**EVA:** The man in the image is sitting down, holding a glass of beer, and making a gesture or a sign with his hand.
**UTA:** The man in the image is sitting down, talking on his cell phone, and holding his hands up while doing so.

Figure 2: Qualitative examples generated by LLaVA models fine-tuned with EVA-02 and UTA.

Table 3: Results on LLaVA-Bench (Liu et al., 2023). The results of CLIP and EVA-02 are obtained by our re-implementation with official checkpoints.

| Method | Model | Conversation | Detail | Reasoning | Overall |
|--------|-------|--------------|--------|-----------|---------|
| CLIP   | B/16  | 74.5         | **69.9** | 90.3    | 78.3    |
| EVA-02 | B/16  | 75.3         | 61.1   | **91.8**  | 76.2    |
| UTA    | B/16  | **80.8**     | 66.2   | 88.8      | **78.8** |
| CLIP   | L/14  | 78.7         | 70.4   | 90.0      | 79.8    |
| EVA-02 | L/14  | 80.4         | 71.6   | 91.1      | 80.6    |
| UTA    | L/14  | **81.4**     | **72.2** | **91.8** | **82.0** |
| EVA-01 | g/14  | 79.9         | **72.2** | 91.0    | 80.8    |
| UTA    | g/14  | **84.1**     | 71.3   | **93.5**  | **83.1** |

### 4.2 MULTI-MODAL RESULTS

The emergent multi-modal capabilities of GPT-4 (OpenAI, 2023) have attracted widespread attention, and there are various re-implementations of such capabilities using open-sourced vision and large language models (Liu et al., 2023; Zhu et al., 2023). We adopt the LLaVA framework and evaluate pre-trained models on the LLaVA-Bench. The results are presented in Tab. 3. Note that all the results are obtained by fixing the vision encoders' parameters, which can directly reflect the representation quality of the pre-trained model. Notably, our model achieves the best results in the overall category. Compared to the original CLIP large model (Radford et al., 2021), we overall obtain an improvement of 2.2% accuracy. Using the same pre-training dataset and iterations, we also outperform EVA-02 (Fang et al., 2023a) for 1.4%. We compare the outputs generated by the two LLaVA models and highlight the difference in Fig. 2. We show that our approach can capture more fine-grained details to produce better answers.

### 4.3 CORE VISION TASK RESULTS

Prior arts (Bao et al., 2021; He et al., 2021) demonstrate that the MIM pre-trained models have superior performance after fine-tuning to downstream tasks, including ImageNet classification, object detection, image segmentation, etc. There are some recent papers (Xie et al., 2023) that show the mask-then-predict objective is the key to such fine-tuning capabilities. In our empirical evaluation, we show that our UTA pre-training also has such capabilities.

We present the results of ImageNet classification in Tab. 4. Compared to recent MIM works (e.g., BEiT v2 (Peng et al., 2022)) which also utilize pre-trained CLIP model for pre-training, we obtain an improvement of ∼2% points after fine-tuning. We can also largely outperform the CLIP model for

Table 4: ImageNet classification and ADE20K segmentation results. ZS and FT denote the zero-shot and fine-tuning top-1 accuracy on ImageNet respectively. † denotes the model after contrastive fine-tuning.

| Method | Model | #Params | ImageNet | | | ADE20K | |
| | | | Input Size | ZS | FT | Input Size | mIoU |
|---|---|---|---|---|---|---|---|
| MAE | B | 86M | 224 | - | 83.6 | 512 | 48.1 |
| BEiT v2 | B | 86M | 224 | - | 85.5 | 512 | 53.1 |
| CLIP | B | 86M | 224 | 68.3 | 85.7 | - | - |
| EVA-02 | B | 86M | 224 | - | 87.4 | 512 | 55.3 |
| UTA | B | 86M | 224 | 76.0 | **87.5** | 512 | **55.6** |
| UTA† | B | 86M | 224 | **77.0** | 87.4 | 512 | 55.1 |
| MAE | L | 304M | 224 | - | 85.9 | 512 | 53.6 |
| BEiT v2 | L | 304M | 224 | - | 87.3 | 512 | 56.7 |
| CLIP | L | 304M | 224 | 74.0 | 88.0 | - | - |
| EVA-02 | L | 304M | 224 | - | 89.0 | 512 | 58.3 |
| UTA | L | 304M | 224 | **78.5** | **89.2** | 512 | **58.8** |
| EVA-CLIP | g | 1011M | 224 | 79.3 | 89.1 | 512 | 57.4 |

Table 5: Object detection and instance segmentation results on COCO and LVIS datasets. † denotes the model after contrastive fine-tuning.

| Method | Model | #Enc. Params | COCO | | LVIS | |
| | | | AP$^{box}$ | AP$^{mask}$ | AP$^{box}$ | AP$^{mask}$ |
|---|---|---|---|---|---|---|
| ViTDet | B | 86M | 54.0 | 46.7 | 43.0 | 38.9 |
| EVA-02 | B | 86M | 55.5 | 47.1 | 47.1 | 41.4 |
| UTA | B | 86M | **55.8** | **47.7** | **49.1** | **43.1** |
| UTA† | B | 86M | 55.6 | 47.5 | 47.9 | 42.2 |
| ViTDet | L | 304M | 57.6 | 50.0 | 49.2 | 44.5 |
| EVA-02 | L | 304M | 58.5 | 50.3 | 55.3 | 48.6 |
| UTA | L | 304M | **58.7** | **50.5** | **55.9** | **49.5** |
| EVA-CLIP | g | 1011M | 59.1 | 51.1 | 56.4 | 51.3 |

both the zero-shot and fine-tuning accuracy. Compared with EVA-02, although we slightly improve the fine-tuning accuracy, we can largely improve the zero-shot accuracy.

We show the results of performing object detection and instance segmentation on COCO and LVIS datasets in Tab. 5. Compared to the MAE pre-training (He et al., 2021), we find our UTA can improve the AP$^{box}$ for more than 1% mAP on COCO and 6% mAP on more challenging LVIS. Additionally, our approach also performs better than EVA-02, which demonstrates 2.0% and 0.6% mAP improvements on LVIS for the base and large models respectively.

## 5 ABLATION STUDIES

In this section, we conduct ablation studies to evaluate the impact of different design choices of our proposed Unmasked Token Alignment (UTA). Unless otherwise specified, we use the ViT-B backbone and pre-train it for 90 epochs on the ImageNet-21K (Deng et al., 2009) dataset.

**Pre-training objectives.** We thoroughly explore the effect of pre-training objectives and show the results in Tab. 6. We also explore combining UTA and MIM by inputting masked and unmasked tokens simultaneously and conducting token alignment for unmasked tokens and feature prediction for masked tokens. We find that UTA performs best on all evaluated benchmarks while requiring the least computation cost. In particular, we find the improvements on LVIS are most significant compared to other approaches. Moreover, we show that combining UTA and MIM can lead to much worse zero-shot accuracy but similar fine-tuning accuracy on ImageNet than using UTA alone. We suspect the training-finetuning inconsistency introduced by the extra [MASK] tokens is more significant when the backbone is fixed for evaluation.

Table 6: The effect of pre-training objectives. FD denotes the re-implementation of the Feature Distillation method (Wei et al., 2022). ZS and FT denote the zero-shot and fine-tuned top-1 accuracy on ImageNet respectively.

| Config | FLOPs | ImageNet | | COCO | | LVIS | | ADE20K |
| | | ZS | FT | $AP^{box}$ | $AP^{mask}$ | $AP^{box}$ | $AP^{mask}$ | mIoU |
|---|---|---|---|---|---|---|---|---|
| FD | 1.0× | 74.7 | 87.2 | 55.2 | 47.0 | 47.9 | 42.2 | 54.7 |
| MIM | 1.0× | - | 86.9 | 54.7 | 46.6 | 46.6 | 41.1 | 54.3 |
| UTA+MIM | 1.0× | 70.7 | 87.2 | 55.4 | 47.1 | 47.7 | 42.0 | 54.8 |
| UTA | 0.6× | **75.0** | **87.3** | **55.7** | **47.4** | **48.9** | **43.1** | **55.4** |

Table 7: The effect of positional embedding. PE denotes w/ or w/o positional embedding during pre-training.

| Method | PE | ImageNet | | COCO | | ADE20K |
| | | ZS | FT | $AP^{box}$ | $AP^{mask}$ | mIoU |
|---|---|---|---|---|---|---|
| MIM | ✗ | - | 85.8 | 50.9 | 43.2 | 51.8 |
| MIM | ✓ | - | 86.9 | 54.7 | 46.6 | 54.3 |
| *Performance gap* | - | -1.1 | -3.8 | -3.4 | -2.5 |
| UTA | ✗ | 73.8 | 86.7 | 53.8 | 45.7 | 53.6 |
| UTA | ✓ | 75.0 | 87.3 | 55.7 | 47.4 | 55.4 |
| *Performance gap* | -1.2 | -0.6 | -1.9 | -1.7 | -1.8 |

**Positional embedding.** Compared to UTA which directly conducts token alignment on unmasked tokens, MIM relies on the unmasked tokens to predict the features of the masked tokens. We speculate that the MIM approach is more susceptible to the influence of positional embedding. We conduct an experiment to remove all the positional embedding in the ViT architecture during pre-training. For fine-tuning, we add the positional embedding back but initialize them with zero to ensure that the initial state of fine-tuning is the same as the last state of pre-training. As shown in Tab. 7, we find that the performance drop of UTA is much smaller compared to MIM. In particular, MIM has 3.8 $AP^{box}$ and 3.4 $AP^{mask}$ performance drop on COCO, while UTA only drops by half of the accuracies.

**Different pre-trained CLIP models.** We study the impact of different pre-trained CLIP models on downstream performance. As shown in Tab. 8, we find that using a stronger CLIP model can lead to better downstream performance. Additionally, we observe that the performance gap was not as significant as on COCO and ADE20K, probably because the classes of those datasets can already be easily classified by CLIP-L/14.

**UTA for pre-training the text encoder.** While we perform UTA to pre-train only the vision encoder by default, we also explore using it to also pre-train a text encoder from scratch. We train a smaller text encoder on DataComp-1B for 1 epoch. Empirically, we only obtain 54.5% zero-shot accuracy after pre-training, which is much lower than using the CLIP text encoder. Thus, we decide to not perform UTA for pre-training the text encoder.

**Mask ratio and mask type.** We examine the effect of the mask ratio and mask type on the final performance. As shown in Tab. 9 (left), we find that using a mask ratio of 0.4 achieves the best computation-performance trade-off. Additionally, using the block-reversed masking performs best on all evaluated datasets.

## 6    RELATED WORKS

**Vision (-Language) Foundation Models.** The Transformer architecture (Vaswani et al., 2017) has rapidly evolved to become a pivotal paradigm in both Computer Vision (CV) and Natural Language Processing (NLP). Models like BERT (Devlin et al., 2019) and the GPT (Floridi & Chiriatti, 2020) series, built upon the Transformer architecture, have exhibited exceptional prowess across various language tasks. Simultaneously, in the field of CV, Vision Transformers (ViTs) (Dosovitskiy et al., 2021) have emerged as potent contenders, gradually displacing CNNs in various downstream vision tasks. Furthermore, the fusion of text and images in a shared embedding space, exemplified by CLIP (Radford et al., 2021), has rendered the Transformer an indispensable tool for versatile uni- and

Table 8: The effect of pre-trained CLIP model.

| CLIP Model | ZS | ImageNet | | COCO | | ADE20K |
| | | ZS | FT | AP$^{box}$ | AP$^{mask}$ | mIoU |
|---|---|---|---|---|---|---|
| CLIP-L/14 | 74.0 | 67.7 | 86.6 | 55.6 | 47.3 | 53.7 |
| EVA-CLIP-g/14 | **79.3** | **75.0** | **87.3** | **55.7** | **47.4** | **55.4** |

Table 9: The effect of mask ratio (left) and mask type (right). Block-R denotes the reversed block-wise masking. We use mask ratio of 0.5 for the mask type ablation.

| Ratio | FLOPs | ImageNet | | COCO | | ADE20K |
| | | ZS | FT | AP$^{box}$ | AP$^{mask}$ | mIoU |
|---|---|---|---|---|---|---|
| 0.0 | 1.0× | 74.7 | 87.2 | 55.2 | 47.0 | 54.7 |
| 0.4 | 0.6× | **75.0** | **87.3** | **55.7** | **47.4** | **55.4** |
| 0.5 | 0.5× | 74.8 | 87.3 | 55.3 | 46.8 | 55.0 |
| 0.7 | 0.3× | 74.0 | 87.0 | 55.0 | 46.6 | 54.8 |

| Mask | ImageNet | | COCO | | ADE20K |
| | ZS | FT | AP$^{box}$ | AP$^{mask}$ | mIoU |
|---|---|---|---|---|---|
| Block | 74.2 | 87.2 | **55.3** | 46.6 | 47.8 |
| Random | 74.7 | 87.2 | 55.1 | 46.4 | 47.7 |
| Block-R | **74.8** | **87.3** | 55.3 | **46.8** | **55.0** |

multi-modal tasks. As training CLIP requires a large amount of computation resources, FLIP (Li et al., 2023b) proposes to mask the visual input tokens to accelerate the training process of CLIP. Recently, large-scale visual pre-training methods based on the Transformer architecture, such as BEiT-3 (Wang et al., 2022a) and EVA (Sun et al., 2023), have continuously pushed the performance boundaries of various downstream visual tasks. In this work, we introduce a simple yet effective large-scale pre-training method for enhancing the multi-modal representations and demonstrate competitive performance on various uni- and multi-modal tasks.

**Masked Image Modeling (MIM).** MIM is a popular pretext task where the vision model learns rich visual representations by conducting reconstruction from corrupted images. Its initial introduction can be traced back to ViT (Dosovitskiy et al., 2021) and iGPT (Chen et al., 2020). Subsequent advancements in the field, exemplified by the notable contributions of BEiT (Bao et al., 2021), MAE (He et al., 2021), and others (Wang et al., 2022b; Liu et al., 2022; Xie et al., 2021), have consistently elevated the performance of the MIM method across diverse downstream tasks. Recent works (Fang et al., 2023b; Peng et al., 2022; Hou et al., 2022; Xiao et al., 2022) have highlighted the utilization of carefully devised reconstruction targets, like the hidden features from a pre-trained CLIP model, which has been shown to facilitate MIM in acquiring superior visual representations. However, these methods rely on the [MASK] tokens to predict the masked features/pixels which introduces the training-finetuning inconsistency. While UMT (Li et al., 2023a) does not use the [MASK] tokens and only processes the unmasked tokens, it focuses on training video models and does not align with the CLIP text model without contrastive fine-tuning. In contrast, our UTA automatically aligns the train-from-scratch ViT model with CLIP text model and enables zero-shot evaluation even without training on image-text pairs.

## 7 CONCLUSION

In this paper, we introduce the Unmasked Token Alignment (UTA) method, which enhances the alignment between vision and language representations by leveraging pre-trained CLIP models. UTA trains a Vision Transformer (ViT) by aligning the unmasked tokens with corresponding visual tokens of a frozen CLIP model. UTA does not suffer from training-finetuning inconsistency and is training-efficient by avoiding using extra [MASK] tokens. The pre-trained ViT model and CLIP text model can be directly applied for zero-shot evaluation even without contrastive training on image-text pairs. Experimental results demonstrate the effectiveness of UTA across various uni- and multi-modal downstream tasks, outperforming existing MIM and CLIP methods.

**Limitations** While the proposed UTA method presents promising results and advantages, it also has some limitations. Firstly, UTA relies on the availability of a pre-trained CLIP model, which may limit its applicability in scenarios where such models are not accessible or suitable. Additionally, although UTA achieves strong zero-shot performance without contrastive fine-tuning, it still benefits from further fine-tuning on large-scale image-text pairs, especially for robustness evaluation. While UTA shows great potential for enhancing multi-modal representations, further research is needed to address these limitations and improve its applicability in a wider range of applications.

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

# A APPENDIX

## A.1 TRAINING DETAILS

**Contrastive fine-tuning on DataComp-1B.** We initialize the model with pre-trained ViT encoder and CLIP text encoder and fix the temperature value in CLIP loss to 0.01. We use a total batch size of 49,152 for fine-tuning. Following Sun et al. (2023), we use LAMB (You et al., 2019) optimizer with peak learning rate of $2 \times 10^{-4}$ and $4 \times 10^{-4}$ for base and large models respectively. We use layer-wise learning rate for fine-tuning and set the decay rate to 0.75 and 0.85 for base and large models respectively. The weight decay is set to 0.05 for all models. We use cosine learning rate schedule and decay the learning rate to 0. We use the prompt provided in CLIP (Radford et al., 2021) for zero-shot evaluation.

**Fine-tuning with LLaVA.** Following Liu et al. (2023), we use a two-stage instruction-tuning procedure for LLaVA model training. **Stage 1: Feature alignment.** At this stage, we train the linear projection layer between the frozen vision encoder and the Large Language Model (LLM) for 1 epoch, utilizing a filtered dataset containing 585K image-text pairs from CC-3M (Sharma et al., 2018). We use AdamW (Loshchilov & Hutter, 2017) optimizer with a learning rate of $2 \times 10^{-3}$. The learning rate is linearly warmed up for the first 150 iterations and decayed to 0 with cosine schedule. We use a batch size of 128 and apply no weight decay. **Stage 2: End-to-end fine-tuning.** We fine-tune the LLaVA model using 158K unique language-image instruction-following dataset for 3 epochs while keeping the vision encoder weights frozen. We use the same optimizer and learning rate schedule as in the first stage except for changing the batch size to 32 and setting the learning rate to $2 \times 10^{-5}$. We do not apply weight decay during this stage either.

**Object detection and segmentation.** Following (Li et al., 2021), we adopt Cascade Mask R-CNN (He et al., 2017; Cai & Vasconcelos, 2019) framework for fine-tuning on COCO (Lin et al., 2014) and LVIS (Gupta et al., 2019). We follow most of the hyper-parameters settings in EVA-02 (Fang et al., 2023a). On COCO, we use batch size of 128 and fine-tune for 60k iterations. We use learning rate of $5 \times 10^{-5}/6 \times 10^{-5}$, drop path rate (Huang et al., 2016) of 0.1/0.4, layer-wise decay rate of 0.7/0.8 for base/large models. On LVIS, we use batch size of 64 and fine-tune for 100k iterations. The learning rate is set to $10^{-4}$. The drop path rate and layer-wise decay rate are the same as those used on COCO. We adopt the UperNet (Xiao et al., 2018) framework for semantic segmentation on ADE20K (Zhou et al., 2017). In particular, we use batch size of 32 and fine-tune for 60k iterations. We use learning rate of $6 \times 10^{-5}/4 \times 10^{-5}$, drop path rate of 0.15/0.2, layer-wise decay rate of 0.85/0.9 for base/large models.

