# OpenReview forum: "Enhancing Vision-Language Model with Unmasked Token Alignment at Scale"
_ICLR.cc/2024/Conference — ICLR 2024 Conference Withdrawn Submission_

### Official Review · Reviewer_SDrz · 2023-10-28

**Soundness:** 2 fair
**Presentation:** 2 fair
**Contribution:** 2 fair
**Rating:** 3
**Confidence:** 4

**Summary:**

This paper claims that crontrastive pretraining from scratch is computationally demanding and masked image modeling will introduce training-finetuning inconsistency problem. Thus, they use the CLIP model as the teacher, to train a vision transformer. When performing distillation, they do not align the masked token as before, but the unmasked token instead. In zero-shot and finetuning experiments, the proposed method gains a large improvement.

**Strengths:**

- The method is described clearly.

- The workload of the experiment is heavy.

**Weaknesses:**

- The biggest weakness is the motivation. Prior works used the masked tokens as the prediction tokens to learn their representation. However, in this paper, they predict the unmasked token instead. Thus, I wonder what information can masked token provide if the model aims to predict the unmasked token. In the former method, they provide the context information with the unmasked tokens. The learning of these former methods makes the context increasingly effective. But in this paper, since unmasked tokens are represented with no difference from the normal tokens, what novel information can the model learn by recovering such unmasked tokens based on similar normal tokens? Or to say the least, the method proposed by the author is similar to the simple distillation process which drops tokens randomly.
- In section 2.2, the author claims that ‘It causes training-finetuning inconsistency and makes the trained ViT unable to perform zero-shot classification without fine-tuning’. In my opinion, this description is not accurate because the trained ViT could be able to transfer to unseen domains no matter whether it is with or without [MASK] token training. So I hope the author can present citations or experimental results to prove this statement.
- In terms of technical novelty, this paper lacks some careful design. Besides, there seems to be a weak connection between the part of the ablation study and motivation, such as **the positional embedding** analysis. Since ICLR is a top-tier conference, this paper also lacks a solid theoretical foundation or forward-looking research direction and should be revised.

**Questions:**

1. In section main result, I only see some statistics, instead of any analysis about increase or decrease. Could you not only present the number which I can see from the figure or table but also give some details about the advantages or disadvantages of the method you proposed?
2. In Tab. 3, could you explain why you useLLaVA-Bench which is often used to test the multimodal instruction ability? And why the UTA with G/14 model did not beat that of L/14 model but exceeded in conversation and reasoning by a large margin?
3. From the results of Tab 4 and Tab 5, it can be seen that the proposed UTA just achieves a very minor improvement compared to EVA-02. How to further justify the superiority of your method in such settings?
4. In ‘UTA for pre-training the textencoder’, there is an interesting result, the same method does help to visionencoder, but not the text encoder. Could you give more discussion? I wonder whether your motivation is convincing and why it does not work in text modality?

**Details Of Ethics Concerns:**

No concerns.

---

> ### Author Response · Authors · 2023-11-17
>
> Thank you for your detailed review. We address the concerns below.
>
> > Q1: Motivation on unmasked token alignment and comparison with MIM.
>
> R1: We share the similar spirit with MIM works (EVA) by using the partial inputs (masked images) to predict the CLIP features obtained by full inputs. However, MIM is a too difficult pretext task which hinders the learning of representations. In addition, using full inputs (without mask) to predict the CLIP features is too easy. Our method balances the learning difficulty by proposing unmasked tokens alignment. Compared to MIM, our method 1) is training-efficient by dropping the masked tokens and 2) can converge more quickly by only aligning the unmasked tokens. We show that UTA can match MIM with only 60% epochs of training. We summarize the results in the table below.
> |Method|Epoch|ImageNet-ZS|ImageNet-FT|COCO-box|COCO-mask|LVIS-box|LVIS-mask|ADE20K|
> |:-:|:-:|:-:|:-:|:-:|:-:|:-:|:-:|:-:|
> |FULL|90|74.7|87.2|55.2|47.0|47.9|42.2|54.7|
> |MIM|90|-|86.9|54.7|46.6|46.6|41.1|54.3|
> MIM|150|-|87.4|55.8|47.7|49.1|43.1|55.3|
> |UTA|90|75.0|87.3|55.7|47.4|48.9|43.1|55.4|
>
> > Q2: The method is similar to the simple distillation process which drops tokens randomly.
>
> R2: Simplicity should be the advantage of our method instead of disadvantage. In our paper, we show that our simple method can already outperform SOTA method on a broad of uni- and multi-modal tasks while being much more training-efficient.
>
> > Q3: The claim “It causes training-finetuning inconsistency and makes the trained ViT unable to perform zero-shot classification without fine-tuning” is not accurate because the trained ViT could be transferred to unseen domains.
>
> R3: The zero-shot classification refers to the evaluation method proposed by CLIP instead of transferring the pre-trained model to unseen domain. As in CLIP, zero-shot evaluation is to use text encoder to encode the classnames, vision encoder to encode images, and match them with cosine similarities. The output features of MIM pre-trained models are not aligned with CLIP text encoder. Therefore, those pre-trained models are unable to perform zero-shot classification without fine-tuning. We also verify the claim in experiments by conducting zero-shot classification with the pre-trained EVA model which uses [MASK] tokens. We only obtain 0.1% accuracy on ImageNet-1K, which is the same as random guess.
>
> > Q4: In terms of technical novelty, this paper lacks some careful design.
>
> R4: Our method is designed to outperform SOTA EVA-CLIP and Open-CLIP with simple modifications. Simplicity should be considered as a part of technical novelty but not a disadvantage.
>
> > Q5: Weak connection between the part of the ablation study (Tab. 7) and motivation.
>
> R5: The ablation study enhances our method by showing that our approach is robust in terms of position embedding. We speculate that MIM with pre-trained CLIP can be seen as a combination of (1) unmasked token alignment as our approach does (2) performing interpolation of the unmasked tokens’ features to predict the masked tokens. So we believe that if we remove the position embedding, i.e., break down the (2) part, MIM will perform worse, which is shown in Table 7.
>
> > Q6: Give some details about the advantages or disadvantages of the method you proposed in our experiments.
>
> R6: Our experiments show that our method has strong performance and is training efficient. Our ablation studies show the impact of different design choices. We present those results with tables and detailed descriptions, which we believe are comprehensive. I hope the reviewer can specify which advantage or disadvantage we need to add. We are willing to include it in the revised version.
>
> > Q7: Why use LLaVA-Bench for testing?
>
> R7: Large Multi-modal Model (LMM) is a highly regarded research direction in computer vision, and the latest LMM are all based on SOTA EVA-CLIP/Open-CLIP models. To have a comprehensive comparison with those SOTA CLIP models, we test our model’s performance on LLaVA-Bench.
>
> > Q8: Why UTA-g/14 model did not beat that of L/14?
>
> R8: Note that the overall performance of UTA-g/14 (83.1) is better than L/14 (82.0). The overall performance averages the performance of the other 3 categories, which can be a more comprehensive indicator of performance.
>
>
> > Q9: Minor improvement compared to EVA-02.
>
> R9: Note that our method performs better than EVA-02 on all tested tasks. We believe the comprehensive improvements over SOTA method are not trivial.
> It is worth noting that our method achieves a 0.5 mIoU improvement on ADE20K, as well as a 0.6 increase in $AP^{box}$ and 0.9 in $AP^{mask}$ on LVIS, which are substantial improvements. Additionally, our method is training-efficient and can converge faster as stated in R1.

---

> > ### Author Response · Authors · 2023-11-17
> >
> > > Q10: Why the method does not work in text modality?
> >
> > R10: The text encoder in CLIP is very small, and the benefits of such pre-training could be limited.
> > Another reason may be related to the modality difference between image and text. The image is spatially redundant. So we can drop a lot of tokens during the pre-training and still obtain good performance even without fine-tuning, which is also shown in MAE or FLIP paper. However, this is not the case for text modality as the information density of text modality is large. Dropping a lot of tokens can lead to large performance drop.

---

> > > ### Author Response · Authors · 2023-11-21
> > >
> > > Dear reviewer, we sincerely appreciate your thoughtful and constructive comments on our paper. We have carefully considered your feedback and made revisions accordingly. Could you inform us if our adjustments have effectively addressed your concerns? We are open to any further discussions.

---

### Official Review · Reviewer_g84W · 2023-10-30

**Soundness:** 3 good
**Presentation:** 3 good
**Contribution:** 2 fair
**Rating:** 3
**Confidence:** 4

**Summary:**

This paper aims to distill the knowledge of a pretrained CLIP model and further enhance its vision-language representations. The core idea is to train a VIT with the masked image modeling objective that aligns the unmaked token embeddings with the CLIP embeddings. After performing the masked image modeling, the VIT and CLIP text encoders are further finetuned with an image-text contrastive loss. The experiments are conducted on a wide range of benchmark datasets and show very promising performance.

**Strengths:**

-It is a good idea to perform unmask token alignment with the pretrained CLIP model due to its efficiency.

-The experiments are quite extensive including many downstream tasks. The method also achieves the SOTA on most of the datasets.

**Weaknesses:**

-Motivation of reducing pretraining costs is not convincing. In particular, the abstract claims that "training CLIP from scratch on noisy web-scale datasets is computationally demanding". Although this is true, this paper does not solve this issue at all because it still relies on a pretrained CLIP model at the first place.

-The performance gains seem to come from the contrastive finetuning rather than the proposed unmasked alignment pretraining. In Table 1, comparing with the CLIP teacher i.e., EVA-CLIP, there is always a performance drop for UTA without finetuning. This is concerning because this paper claims the unmasked alignment pretraining as one of the main contributions.

-One important baseline is missing. Since finetuning is very effective for zero-shot image classification, this paper should also compare with the CLIP teacher i.e., EVA-CLIP,  that is also finetuned on the same dataset.

-Improving the VIT efficiency by dropping masked tokens has been done in [A].  This paper fails to cite this important reference paper and claims it to be something new.

[A] Li et al., Scaling Language-Image Pre-training via Masking. CVPR 2023

-The improvement in Table 4 & 5 (detection and segmentation) is marginal (< 1%). This is also concerning. The method seems to be limited to image-level prediction tasks.

**Questions:**

The authors are highly encouraged to address my questions mentioned in the weakness. In addition, I have the following questions.

-Is the CLIP teacher-model always the giant EVA-CLIP?

-It would be good to provide the CLIP teach results in Table 3, 4, & 5.

-This paper says that it is following previous works to perform the second-stage contrastive finetuning without proving the reference. Please provide the reference.

-Is finetuning helpful in Table 4 & 5?

-Is masking strategy applied in the finetuning stage?

---

> ### Author Response · Authors · 2023-11-17
>
> Thank you for your detailed review. We address the concerns below.
> > Q1: Motivation of reducing pre-training cost is not convincing as UTA relies on a pre-trained CLIP model at the first place.
>
> R1: Even with a pre-trained CLIP model, our total cost is still lower than EVA-CLIP or Open-CLIP. Compared to EVA-02-CLIP-B, we use the same training schedule but require much less training cost. Note that EVA-02-CLIP-B is obtained by contrastive fine-tuning from EVA-02-B. In the pre-training stage, we use the same dataset (ImageNet-21K), teacher model (EVA-01-CLIP-g), and training epochs (150) as in EVA-02-B.
> However, UTA significantly reduces training FLOPs by 40% by not processing masked tokens, a key distinction from the approach of EVA-02-B.
> For the contrastive fine-tuning, we require fewer image-text pairs (2B vs. 8B in EVA-B-CLIP) but achieve better results (77.0 acc. vs. 74.7 acc.). We break down the training samples and show the performance in the table below.
>
> |Method|Pre-training|Contrastive fine-tuning|Zero-shot acc.|
> |:-:|:-:|:-:|:-:|
> |EVA-02|2B|8B|74.7|
> |UTA|2B|0B|76.0|
> |UTA|2B|2B|77.0|
>
> Compared to Open-CLIP-g (34B pairs, 78.5 acc.), our results are UTA-g (0B pair, 79.3 acc.) and UTA-CLIP-g (2B pairs, 81.5 acc.). Considering that the teacher CLIP model uses 11B pairs, our method still requires fewer image-text pairs and obtains better results than CLIP models that train from scratch.
>
> > Q2: Performance gains come from contrastive fine-tuning.
>
> R2: The SOTA EVA-CLIP also used contrastive fine-tuning. Taking the EVA-02-CLIP-B as an example, they first use a ViT-B model to perform masked image modeling with EVA-01-CLIP-g as the teacher, and then use the pre-trained ViT-B model to perform contrastive fine-tuning with 8B image-text pairs, which obtains 74.7 zero-shot accuracy. In comparison, we use the same teacher model to performance UTA pre-training, and further perform contrastive fine-tuning with 2B image-text pairs. Our pre-trained model obtains 76.0 zero-shot accuracy. Our fine-tuned model obtains 77.0 zero-shot accuracy. The training setting is a little complicated. However, we use the same setting in EVA-02 to ensure fair comparison. Note that EVA-02 is an improved version of original EVA, and we focus on the comparison with EVA-02 in our paper. We updated Tables 1 and 2 in our revised version for clarification.
>
> > Q3: Missing baseline that fine-tunes the EVA-CLIP on the same dataset.
>
> R3: As stated in R1 and R2, the results of EVA-CLIP are also obtained by extensive contrastive fine-tuning.
>
> > Q4: Cite FLIP.
>
> R4: Thanks for the suggestion. FLIP is cited in the revised version.
>
> > Q5: Is the CLIP teacher-model always the giant EVA-CLIP?
>
> R5: By default, we use giant EVA-CLIP as the teacher model to align the setting in EVA-02. We ablate the CLIP teacher model of different sizes in Tab.8, we find that using stronger CLIP model can largely improve the results on classification and semantic segmentation, while this is not the case for object detection or instance segmentation.
>
> > Q6: Provide the CLIP teacher results in Table 3, 4, & 5.
>
> R6: Thanks for the suggestion. The results are added in the revised version.
>
> > Q7: Provide reference that uses second-stage contrastive fine-tuning.
>
> R7: As stated in R1 and R2, the EVA-CLIP utilized second-stage contrastive fine-tuning.
>
> > Q8: Is fine-tuning helpful in Table 4 & 5?
>
> R8: Thanks for the suggestion. We tried such experiments and found that contrastive fine-tuning is not helpful for those tasks.  The conclusion is similar to other MIM works that the task-agnostic pre-training is better than task-specialized models when fine-tuning on other tasks. We summarize the results below and have added the results in our revised version.
>
> |Method|ImageNet-ZS|ImageNet-FT|COCO-box|COCO-mask|LVIS-box|LVIS-mask|ADE20K|
> |:-:|:-:|:-:|:-:|:-:|:-:|:-:|:-:|
> |UTA-B|76.0|87.5|55.8|47.7|49.1|43.1|55.6|
> |UTA-CLIP-B|77.0|87.4|55.6|47.5|47.9|42.2|55.1|
>
> > Q9: Is masking strategy applied in the fine-tuning stage.
>
> R9: No.

---

> > ### Author Response · Authors · 2023-11-21
> >
> > Dear reviewer, we sincerely appreciate your thoughtful and constructive comments on our paper. We have carefully considered your feedback and made revisions accordingly. Could you inform us if our adjustments have effectively addressed your concerns? We are open to any further discussions.

---

### Official Review · Reviewer_4GX4 · 2023-11-01

**Soundness:** 3 good
**Presentation:** 3 good
**Contribution:** 3 good
**Rating:** 6
**Confidence:** 3

**Summary:**

This paper propose a Unmasked Token Alignment (UTA) strategy to improve the performance of pre-trained ViT. It achieves higher performance than CLIP.

**Strengths:**

-The proposed method is a universal strategy to improve the learned representation. The learned features can be well-used on various downstream tasks.

-Compared with MAE, it shows significantly higher performance on ImageNet. It is interesting to discuss which pre-trained strategy is better.

**Weaknesses:**

-Will the new module bring extra training cost?

-It may be unfair to directly compare UTA and MAE, as UTA uses an extra tearcher but MAE is only trained with itself.

**Questions:**

See the weakness.

---

> ### Author Response · Authors · 2023-11-17
>
> Thank you for your detailed review. We address the concerns below.
> > Q1: Will the new module bring extra training cost?
>
> R1: Our overall training cost is lower than SOTA method EVA-02. As shown below (Table 6 in the paper), we achieve stronger results on all tasks with fewer training FLOPs.
> |Method|FLOPs|ImageNet-ZS|ImageNet-FT|COCO-box|COCO-mask|LVIS-box|LVIS-mask|ADE20K|
> |:------:|:------:|:------:|:------:|:------:|:------:|:------:|:------:|:------:|
> |EVA-02|1x|-|86.9|54.7|46.6|46.6|41.1|54.3|
> |UTA|0.6x|75.0|87.3|55.7|47.4|48.9|43.1|55.4|
>
> > Q2: Unfair comparison with MAE as UTA uses extra teacher.
>
> R2: It is important to note that more advanced variants of MAE, such as BEiT-v2, EVA, and EVA-02, also utilize additional teachers in their training methodologies. In our study, we explicitly address this concern by providing a comprehensive comparison with these advanced variants in Tables 4 and 5.

---

> > ### Author Response · Authors · 2023-11-21
> >
> > Dear reviewer, we sincerely appreciate your thoughtful and constructive comments on our paper. We have carefully considered your feedback and made revisions accordingly. Could you inform us if our adjustments have effectively addressed your concerns? We are open to any further discussions.

---

### Official Review · Reviewer_WgGM · 2023-11-09

**Soundness:** 4 excellent
**Presentation:** 3 good
**Contribution:** 4 excellent
**Rating:** 3
**Confidence:** 4

**Summary:**

This paper proposes an approach to train a ViT with the target of "aligning" with existing CLIP visual tokens. Specifically, during training, part of the image tokens are masked out and the learning target is to "align" the resulting embedding with the unmasked portion of the CLIP visual tokens. After the pre-training stage, the model can be further fine-tuned on image-text pair data to further enhance its cross-modal capability. The authors conduct extensive experiments on benchmark datasets that show good performance on a number of zero-shot image classification, vision-only and vision-language tasks.

**Strengths:**

- The approach is very intuitive and technically sound, with good reproducibility.
- The paper is well written, with clear motivation, approach and detailed experimental results.

**Weaknesses:**

- Contribution is small. UTA is essentially a variant of the popular Feature Distillation (FD) approach. Ablation study in table 6 indeed show that the performance of UTA is only slightly better than FD.
- Zero-shot performance on ImageNet zero-shot is a bit unfair when comparing UTA against open-CLIP / EVA-CLIP, as the former use ImageNet-21k for training (although without labels). A fairer experiments in my opinion would be pre-training UTA on a random set of unlabeled web images.

**Questions:**

see above

---

> ### Author Response · Authors · 2023-11-17
>
> Thank you for your detailed review. We address the concerns below.
>
> > Q1: Contribution and performance gap are small compared to Feature Distillation.
>
> R1: We appreciate the reviewer's attention to the comparison between our proposed method (UTA) and Feature Distillation (FD).
> 1) Note that the results of FD in Table 6 are our revised version by incorporating negative cosine loss. To ensure a fair comparison, we employed the same dataset and architecture for training. The original FD uses smooth L1 loss for training, and therefore the pre-trained vision encoder cannot be used for zero-shot evaluation because the embeddings of pre-trained vision encoder and CLIP text encoder are not aligned in the normalized embedding space. In comparison, the pre-trained vision encoder of UTA can be directly applied for zero-shot evaluation and obtain decent zero-shot performance after the unmasked alignment.
> 2) Our method outperforms FD across all tasks while reducing the pre-training FLOPs by a significant 40%. Specifically, on COCO, our improvements include 0.5 $AP^{box}$ and 0.4 $AP^{mask}$. On more challenging LVIS, we observe improvements of 1.0 $AP^{box}$ and 0.9 $AP^{mask}$. Additionally, on ADE20K, we improve FD by 0.7 mIoU. We emphasize that these performance enhancements are substantial, considering our utilization of a strong baseline setting. While acknowledging that the improvement on ImageNet is relatively modest, we attribute this observation to the comprehensiveness of the training dataset (ImageNet-21K), which effectively covers ImageNet.
>
>
> > Q2: Unfair to compare UTA against Open-CLIP / EVA-CLIP as UTA uses ImageNet-21K.
>
> R2: The SOTA EVA-CLIP also used ImageNet-21K for pre-training. To ensure strictly fair comparison with SOTA EVA-CLIP, we use the same setting for pre-training, including dataset and architecture. Note that the EVA-CLIP models are fine-tuned from EVA-02 models, which also use ImageNet-21K for pre-training. As the settings of Open-CLIP and EVA-CLIP are very different, we choose the stronger EVA-CLIP as the baseline setting.

---

> > ### Author Response · Authors · 2023-11-21
> >
> > Dear reviewer, we sincerely appreciate your thoughtful and constructive comments on our paper. We have carefully considered your feedback and made revisions accordingly. Could you inform us if our adjustments have effectively addressed your concerns? We are open to any further discussions.